# Fast Volumetric Feedback under Microscope by Temporally Coded Exposure Camera

**DOI:** 10.3390/s19071606

**Published:** 2019-04-03

**Authors:** Kazuki Yamato, Toshihiko Yamashita, Hiroyuki Chiba, Hiromasa Oku

**Affiliations:** Graduate School of Science and Technology, Gunma University, 1-5-1, Tenjin-cho, Kiryu 376-8515, Japan; t13304128@gunma-u.ac.jp (T.Y.); t181d040@gunma-u.ac.jp (H.C.); h.oku@gunma-u.ac.jp (H.O.)

**Keywords:** high-speed measurement, three-dimensional information, feedback control

## Abstract

We developed a temporally coded exposure (TeCE) camera that can cope with high-speed focus variations of a tunable acoustic gradient index (TAG) lens. The TeCE camera can execute a very short exposure multiple times at an arbitrary timing during one shot. Furthermore, by accumulating the photoelectrons generated by each exposure, it is possible to maintain the brightness even with a short exposure time. By synchronously driving the TeCE camera and the TAG lens, different focal planes of an observation target can be acquired at high speed. As a result, high-speed three-dimensional measurement becomes possible, and this can be used for feedback of three-dimensional information. In the work described in this paper, we conducted a focus tracking experiment to evaluate the feedback performance of the TeCE camera. From the experimental results, we confirmed the feedback capability of the TeCE camera.

## 1. Introduction

In the fields of biology and microrobotics, three-dimensional information about the object to be observed or manipulated is required. For this reason, there has been extensive research on three-dimensional measurement in microscopic regions [1,2,3,4,5]. However, since the visual field of a microscope becomes more narrow as the magnification of the objective lens increases, it is difficult to grasp an object within the visual field when the object moves about. Therefore, high-speed three-dimensional measurement of an object under a microscope is an important research subject.

Among robots requiring three-dimensional information, surgical robots [6] perform feedback control based on the three-dimensional structure of the object. If the feedback has a delay, causing the three-dimensional information to be acquired at a low sampling rate, feedback control cannot be performed with satisfactory performance. Therefore, in the robotics field, it is important to measure three-dimensional information at a high sampling rate without delay, especially in servo control, where the feedback period is preferably 1 ms or less [7].

A focus scanning system combining an optical microscope and a Z scanner [8,9,10,11], a digital holographic microscope [3,12,13], and a light field microscope [14] have been proposed as practical examples of systems requiring three-dimensional measurement by a microscope. The digital holographic microscope and the light field microscope are not suitable for high-speed three-dimensional measurement since these systems require complex calculations for three-dimensional reconstruction. In our study, we focused on focus scanning. Focus scanning is a method used for acquiring an image while scanning the focal position in the direction of the optical axis (*Z*-axis), and three-dimensional reconstruction is performed based on the series of images acquired at the individual focal positions. This three-dimensional reconstruction can be performed with comparatively simple processing, so it is suitable for high-speed three-dimensional measurements like those used in feedback control. However, the operating speed of existing Z scanners is about 50 Hz at most, which is not sufficient for high-speed measurement.

In recent years, liquid lenses capable of varying the focal position at high speed, called tunable acoustic gradient index (TAG) lenses, have been developed [15]. A TAG lens vibrates the liquid enclosed therein and uses the density distribution of the liquid generated by the vibration as a lens. During vibration, the refractive power is changed by utilizing a natural vibration mode having point symmetry with respect to the optical axis. The TAG lens functions as a convex lens when the center of the liquid is dense and a concave lens when it is sparse. The natural vibration frequency of the liquid is extremely high, typically 50 to 500 kHz, and the focal position is also made to fluctuate at 50 to 500 kHz. However, since the TAG lens utilizes the natural vibration of the liquid, the lens cannot be stopped at a specific focal position. In addition, while the focus variation is performed at high speed, the fluctuation period is very short. Therefore, in order to extract information about a specific focal position, it is necessary to perform an extremely short exposure of 0.1 to 1 μs at the moment the lens reaches that focal position. However, general image sensors do not support such a short exposure, and an image taken with a short exposure becomes very dark.

To solve this problem, a method was proposed to synchronize RGB strobe illumination with the focus fluctuation of the TAG lens [16,17]. This method realizes the same effect as short-time exposure by illuminating the object only when the TAG lens reaches a specific focal position. Furthermore, by causing the strobe to emit illumination of each color (RGB) at different focal positions, different focal plane images can be obtained for each color component of the acquired image. Thus, it is possible to acquire at most three focal planes in one shot using a general RGB camera. However, there is a limit to the speed at which the strobe can be driven.

In order to solve these problems, temporally coded exposure (TeCE) cameras were developed [18]. TeCE cameras can perform exposure multiple times at a given timing during one shooting, and photoelectrons generated by each exposure can be accumulated. In other words, it is considered that a specific focal plane can be extracted by synchronizing the focus variation of the TAG lens and the exposure timing of the TeCE camera. In addition, by executing short-time exposure multiple times during one shooting, the brightness of the image in shooting by short-time exposure is compensated. Compared to the method using color strobe lights, a TeCE camera is more practical because there are no restrictions in terms of illumination, and image acquisition is possible simply by exposure control. In order to evaluate the performance of the TeCE camera, we evaluated four kinds of the optical characteristics of the TeCE camera: (i) Change in brightness depending on exposure time; (ii) depth of field; (iii) change in focal position, and (iv) resolution. Furthermore, an experiment involving feedback of three-dimensional information was performed. From the results, the feedback control capability of the TeCE camera was confirmed.

## 2. Synchronous Driving of TeCE Camera and TAG Lens

In this paper, we propose high-speed volumetric feedback using a TeCE camera.

First, we will describe the TeCE camera. TeCE cameras are high-speed cameras using an improved exposure method in order to extract information at a specific focal plane from a TAG lens. TeCE cameras have two main features. One is that a short-time exposure can be performed multiple times at an arbitrary timing. In general high-speed cameras, while one frame is being shot, exposure is performed in less than one shooting time. For example, when photographing at 100 fps, the exposure time is 10 ms or less. TeCE cameras can execute microsecond exposure multiple times within the shooting time of one frame. Further, the exposure timing can be arbitrarily set as long as it is within the imaging time of one frame. The other feature is that photoelectrons generated by multiple exposures can be accumulated within one frame. As a result, it is possible to compensate for the brightness of the captured image by performing exposure multiple times, even with an exposure time on the order of microseconds.

Next, synchronous driving of the TeCE camera and the TAG lens will be described. Figure 1 shows a schematic diagram of synchronous driving of the TeCE camera and the TAG lens. As shown in the figure, the TAG lens can change its focal length at high speed. However, since it cannot be fixed to an arbitrary focal length, in order to photograph a specific focal plane, it is necessary to perform exposure with an exposure time of microsecond order. This problem can be solved by exploiting the characteristics of the TeCE camera. The TeCE camera can execute microsecond-order exposure at arbitrary timing multiple times within one frame time. As shown in the figure, at the moment the TAG lens reaches a specific focal length, exposure by the TeCE camera is executed. The TeCE camera performs exposure every time the TAG lens reaches this specific focal length within one frame time and accumulates the photoelectrons generated by each exposure in a capacitor so that a specific focal plane can be acquired as an image. In the next frame, it is possible to acquire a focal plane different from the focal plane acquired in the previous frame by performing exposure with a focal length different from that in the previous frame. Therefore, it is possible to acquire different focal planes for each frame by synchronous driving of the TeCE camera and the TAG lens.

Actually, the exposure timing of the TeCE camera is determined by using the trigger signal output from the TAG lens. The trigger signal is output when the focal length of the TAG lens is maximum. Since the period with which the focal length fluctuates is known, the time from the output of the trigger signal to the specific focal length can be calculated. That is, the exposure signal is delayed from the generation of the trigger signal until the TAG lens reaches a specific focal length and is then input to the camera. As a result, the TeCE camera performs exposure at the moment the TAG lens reaches a specific focal length.

The TeCE camera can extract a specific focal plane from the TAG lens simply by controlling the exposure timing. Therefore, unlike a system in which the light source is controlled, the TeCE camera can acquire volumetric information about the target using normal lighting.

## 3. Experiment

We constructed an experimental system for three-dimensional measurement of an object under a microscope by combining the TeCE camera and the TAG lens. We used this experimental system to evaluate the performance of the TeCE camera and the synchronous driving of the TeCE camera and the TAG lens shown in Figure 1. In addition, we evaluated the feedback performance of the TeCE camera by performing a focus tracking experiment.

### 3.1. Construction of Experimental System

As shown in Figure 2, the experimental system consisted of a microscope, a relay lens system, a TAG lens, a field programmable gate array (FPGA), and a TeCE camera. Figure 3a is a photograph of the experimental system actually constructed. Figure 3b shows a photograph of the TeCE camera and the relay lens system connected to a microscope. As shown in Figure 3c, the TAG lens was part of the relay lens system. That is, the relay lens system served as a jig for connecting the TAG lens to the camera port of the microscope. The FPGA outputted an exposure control signal to the TeCE camera using a trigger signal output from the TAG lens and a frame switching signal output from the TeCE camera. The frame switching signal of the TeCE camera was output when shifting from the frame being photographed to the next frame, and this signal was output each time the frame was switched. For example, when shooting at a speed of *n* fps, the frame switching signal is output every $\frac{1}{n}$ s. When shooting different focal planes in each frame, the FPGA changed the exposure timing from the output of the frame switching signal. In the experimental system, the microscope was IX71 (Olympus, Tokyo, Japan), and the objective lens was UPLANSAPO 10X NA 0.4 (Olympus) with a magnification of 10-times. We used MAX 10 (Altera, San Jose, CA, USA) for the FPGA and TAG Lens 2.5 β (TAG Optics, Princeton, NJ, USA) for the TAG lens. Three vibration modes of 69, 189, and 311 kHz could be selected for the TAG lens used in the experiment. Although higher refractive power can be obtained with higher-frequency vibration modes, one vibration period would become shorter. Therefore, the vibration mode was set to 69 kHz, and the amplitude of the vibration was set to 0.5 dioptor so that a range of focal positions suitable for the objective lens used in the experiment could be obtained. The vibration of the TAG lens had a positive side and a negative side, and the refractive power between the extreme value on the positive side and the extreme value on the negative side was 1 dioptor.

### 3.2. Evaluation of Optical Characteristics of TeCE Camera

We evaluated the performance of the TeCE camera prototype described in the previous section. The performance aspects that we evaluated were as follows: (i) Change in brightness depending on exposure time; (ii) depth of field; (iii) change in focal position; and (iv) resolution. As observation targets, for (i) to (iii) we used a Ronchi ruling glass slide, and for (iv) we used a resolution chart. The TeCE camera was configured to have a frame size of 640 × 480 pixels and a frame rate of 1000 fps. Also, the single exposure time was 1.3 μs. Details of the evaluation of these four aspects of the system performance are given below.

#### 3.2.1. Change in Brightness Depending on Exposure Time

We evaluated the change in brightness of the image acquired by the TeCE camera, depending on the number of exposures executed while taking one frame. In evaluating this aspect, in order to evaluate only the brightness of the image, the TAG lens was not driven. Figure 4 shows the evaluation results for (i). As shown in the figure, it was confirmed that the image became brighter as the number of exposures increased. This is due to the accumulation of photoelectrons generated by each exposure, which is a characteristic of the TeCE camera. In other words, the image that was exposed 50 times during one shot was formed by accumulating photoelectrons for 50 exposures. The shooting time of one frame was 1 ms, and the focal vibration of the TAG lens took about 14.5 μs for one vibration. Assuming that exposure is performed once for each vibration of the TAG lens, the number of exposures is 60 times or more. That is, under this condition, it would be possible to obtain an image brighter than the image formed by 50 exposures shown in the figure.

#### 3.2.2. Depth of Field of Experimental System

The procedure for measuring the depth of field will be explained with reference to Figure 5. First, the Ronchi ruling was set at an angle of 11.02 degrees, the objective lens was adjusted so that the center of the image was in focus, and an image was captured. Note that the TAG lens was driven and the TeCE camera was set so as to perform exposure simultaneously with the generation of the trigger signal from the TAG lens. Next, the contrast of an arbitrary row in the captured image was calculated by
(1)$$C=\frac{I_{max}-I_{min}}{I_{max}+I_{min}},$$
where $I_{max}$ and $I_{min}$ indicate the maximum pixel value and the minimum pixel value of a pixel group consisting of eight pixels. After that, a contrast profile was created from the calculation results, and the full width at half maximum was obtained from the profile. In this study, we used the full width at half maximum as a measure of the depth of field. The above process was carried out by calculating the full width at half maximum while changing the exposure timing of the TeCE camera. Figure 5 shows the measurement results for the depth of field. From the figure, it can be confirmed that the same degree of depth information could be acquired at any exposure timing.

#### 3.2.3. Change in Focal Position Due to Focal Vibration of the TAG Lens

The evaluation method of change in focal position of the experimental system is explained as follows. First, the objective lens was moved so as to focus on the object to be acquired. At this time, the TAG lens was driven, and the TeCE camera was set so as to be exposed simultaneously with the generation of the trigger signal from the TAG lens. Next, the exposure timing of the TeCE camera was changed by 1 μs at a time. As a result of changing the exposure timing, the focal position was changed, so the object to be acquired appeared to be out of focus. The objective lens was moved to focus on the object again. Note that the movement amount of the objective lens was measured, and this was regarded as the change of the focal position. The above procedure was performed while changing the exposure timing. In addition, a simplified model of the experimental system was constructed by optical CAD, and the change of focal position calculated by optical CAD was also measured as an ideal value. Figure 6 shows the measurement results. It can be confirmed from the figure that the focal position changed by changing the exposure timing, although there was a slight difference in the peak between the ideal value (7 μs) and the measured value (6 μs). The reason for the slight difference is that it is visually judged whether the objective lens is in focus or not.

#### 3.2.4. Resolution

To evaluate the resolution, a resolution chart was used as an observation target. The procedure for measuring the resolution was as follows. First, like the other measurements of aspects (ii) and (iii), the TAG lens was driven, and the exposure timing of the TeCE camera was set to be simultaneous with the generation of the trigger signal from the TAG lens. Next, the resolution chart shown in Figure 7a was acquired, and the resolution was obtained based on the acquired image and the resolution table, as shown in Table 1. At this time, criteria for judging whether or not the resolution chart was focused were visually checked. The above procedure was performed while changing the exposure timing. Figure 7b shows the measurement results. It can be confirmed from the figure that, by driving the TAG lens, the resolution was lower than that before the TAG lens was driven. However, it was possible to confirm the change in resolution by the change in exposure timing. In particular, when the exposure timing was 5, 6, and 12 μs, the resolution was high. It is considered that the exposure was performed when the refractive power of the TAG lens was large. On the other hand, since the resolution was low when the exposure timing was 1, 2, 3, 8, 9, and 10 μs, it is considered that the exposure was performed when the refractive power of the TAG lens was small. From the above results, it can be confirmed that the resolution changed due to the focus variation of the TAG lens.

In addition to the above evaluations, different focal planes were acquired using the experimental system. Figure 8 shows the acquisition results. In addition, as the object to be photographed, a Ronchi ruling was installed at an angle of 10.41 degrees. The reason for this was to make it easy to confirm that different focal planes could be acquired. From the figure, it can be confirmed that different focal planes were acquired by changing the exposure timing. To demonstrate an ability to focus on biological samples with TeCE camera, two-celled embryo of *Caenorhabditis elegans* were measured by the experimental system. Figure 9a,b were images taken at different exposure timings. In other words, Figure 9a,b represented different focal planes. As shown in Figure 9a, the edge or two-celled embryo was generally clear. In Figure 9b, the lower half of the two-cell embryo was sharp but the upper half was blurred. As a result, it was confirmed that Figure 9a,b were the different focal planes each other. Therefore, the focusing ability of the TeCE camera was valid for biological samples.

### 3.3. Focus Tracking

In order to evaluate the fast feedback capability of the TeCE camera, a focus tracking experiment was conducted. Since the experimental system constructed in this paper acquired only two focal planes, it was difficult to acquire volumetric information of a thick object. Therefore, in this experiment, we assumed a thin object with a distinctive texture to be tracked, and we tracked the Ronchi ruling.

The system configuration for the focus tracking experiment consisted of the experimental system, a PC for image processing, a PC for stage control, and an automatic stage, as shown in Figure 10. The flow of focus tracking was as follows. First, two focal planes to be observed were acquired by the experimental system. Next, the image processing PC calculated the target depth information from the two acquired focal planes. Thereafter, the calculated depth information was transferred to the control PC, and the control PC controlled the automatic stage according to the depth information. By repeating the above process at high speed, it was possible to observe the target while keeping the target at a constant depth.

The method of calculating the depth information from two focal planes will be described briefly here. The calculation algorithm that we used was the one described in [17]. Let the focal plane photographed with the TeCE camera be $I_{i}$. Here, *i* is the index of the focal plane. For the sake of convenience, $I_{0}$ is the surface whose focal length is close to the objective lens, and $I_{1}$ is the far surface. First, a gradient $G_{i}$ of each focal plane is obtained as follows:(2)$$\begin{array}{cc} G_{i}\left(x,y\right) & =\frac{1}{2}\{\left|I_{i}\left(x,y\right)-I_{i}\left(x+m,y\right)\right|\\ & +\left|I_{i}\left(x,y\right)-I_{i}\left(x,y+m\right)\right|\}, \end{array}$$
where $I_{i}\left(x,y\right)$ represents the pixel values of position (x,y) in $I_{i}$, and *m* indicates the distance between two pixels. Next, $G_{i}$ is binarized with a threshold value *T*, i.e., if $G_{i}\left(x,y\right)>T$, $B_{i}\left(x,y\right)=1$, otherwise, $B_{i}\left(x,y\right)=0$, where $B_{i}\left(x,y\right)$ denotes the binarized pixels of $G_{i}\left(x,y\right)$. By binarizing the gradient, an area with high contrast is represented by white pixels, and other areas are represented by black pixels. Next, the area $M_{i}$ of white pixels on each focal plane is obtained as follows:(3)$$M_{i}=\sum_{x=1}^{X}\sum_{y=1}^{Y}B_{i}\left(x,y\right),$$
where *X* and *Y* denote the horizontal and vertical sizes of the image, respectively. Let *z* be the depth position of the observation target, let *z* = 0 if the object is in focus at focal plane $I_{0}$, and *z* = 1 if the subject is in focus at focal plane $I_{1}$. The depth position *z* of the target can be expressed by the following equation:(4)$$z=\frac{M_{1}}{M_{0}+M_{1}}.$$
The goal of focus tracking is to keep the target at *z* = 0.5, which is the depth in the middle of the two focal planes. Therefore, it is necessary to transfer *z* calculated from the two focal planes to the control PC and control the stage so that *z* = 0.5. However, since *z* is a fractional value representing the position of the object between the two focal planes, it is necessary to obtain the actual movement amount $z^{\prime }$. $z^{\prime }$ is obtained by the following equation:(5)$$z^{\prime }=\alpha \left(z-\beta \right),$$
where α is a feedback gain that was set to 100, which was determined experimentally, and β is a tracking goal that was set to 0.5, as described above. The parameters were determined experimentally.

The experimental conditions for focus tracking will be described here. The focus variation frequency of the TAG lens was 69 kHz, and the resolution of the focal plane obtained by the TeCE camera was 400×400. The TeCE camera frame rate was set to 1000 fps. The exposure timing of the TeCE camera was exposed simultaneously with the generation of the trigger signal from the TAG lens at $I_{0}$, and $I_{1}$ was exposed 6 μs after the generation of the trigger signal. Each exposure time was 1.3 μs. In order to calculate the depth information $z^{\prime }$ necessary for focus tracking, acquisition of two focal planes is required. Since the acquisition time of one focal plane was 1 ms, the depth information was acquired every 2 ms. Therefore, the feedback rate in this experimental setup was 2 ms. Also, the calculation time of the depth information was measured, which was 0.96 ms per frame, and the standard deviation was 0.43. Shared memory was used for transferring the depth information to the control PC. The accuracy of the automatic stage used in this experiment was 0.25 μm and the control cycle time was 1 ms. Figure 11a shows the automatic stage step response. PD control was used to control the automatic stage by the control PC.

The purpose of the focus tracking experiment was to evaluate the feedback performance of the TeCE camera. The base of the automatic stage used in this experiment had a manual lever that could be raised and lowered. Even if this manual lever is operated, feedback can be realized so long as the automatic stage can be controlled to restore the focal position. In other words, it was evaluated whether focusing could be continued with respect to the operation of the manual lever. The displacement of the manual lever operation was measured using a laser displacement meter.

Figure 11b,c show the experimental results. From the figure, since the actual position of the stage and the command value to the stage are plotted at approximately the same position, it can be confirmed that the stage was operating according to the command value. On the other hand, the displacement of the base by the manual lever was in the opposite direction to the displacement of the automatic stage. This was because the focal position changed due to the manual displacement, and it was considered that the automatic stage was operated so as to cancel the displacement by manual operation. Therefore, it was confirmed that focus tracking was achieved by feedback of depth information created from two focal planes.

## 4. Conclusions

In this paper, we propose a prototype system for three-dimensional measurement of minute regions at high speed by synchronously driving a TeCE camera and a TAG lens, and we evaluated the system experimentally. In the proposed synchronous drive method, the exposure timing of the TeCE camera is determined according to the trigger signal of the TAG lens, and a specific focal plane is extracted. The TeCE camera can perform a very short exposure, and exposures that occur during a single shot can be integrated in a capacitor. Therefore, even in a very short exposure for extracting a specific focal plane from the TAG lens, the brightness of the acquired image can be maintained. We conducted the experiments in order to confirm the effectiveness of TeCE camera. The experimental result of the focus tracking showed that the feedback performance of TeCE camera was 500 Hz. From this result, it was confirmed that TeCE camera could achieve the high-speed three-dimensional measurement.

## Figures and Tables

**Figure 1 sensors-19-01606-f001:**
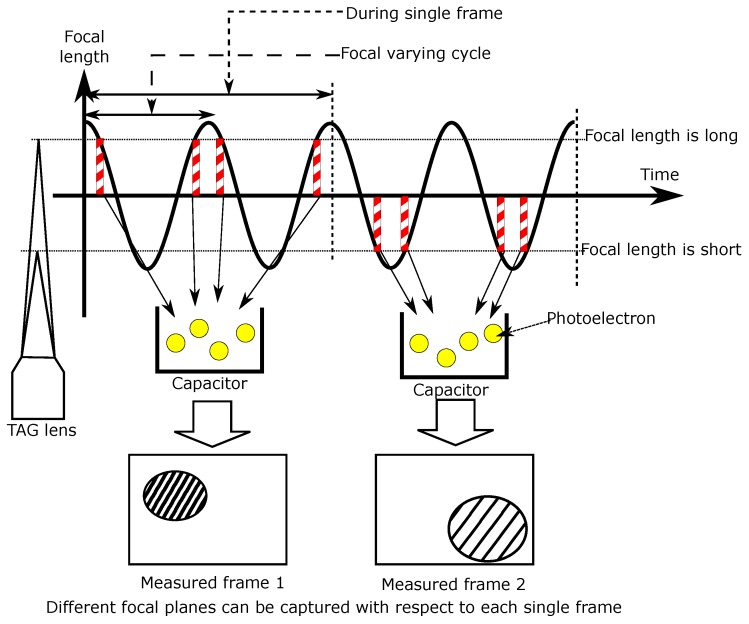
Schematic diagram of synchronous driving of the temporally coded exposure (TeCE) camera and tunable acoustic gradient index (TAG) lens. In measuring frame 1, four short exposures are carried out when the focal length of the TAG lens is long, and the photoelectrons generated by the four exposures are accumulated. Frame 2 is obtained in the same way as frame 1, but the focal length for the exposures is different. If the focal length is different, the focal plane is also different. Therefore, frame 1 and frame 2 capture different focal planes.

**Figure 2 sensors-19-01606-f002:**
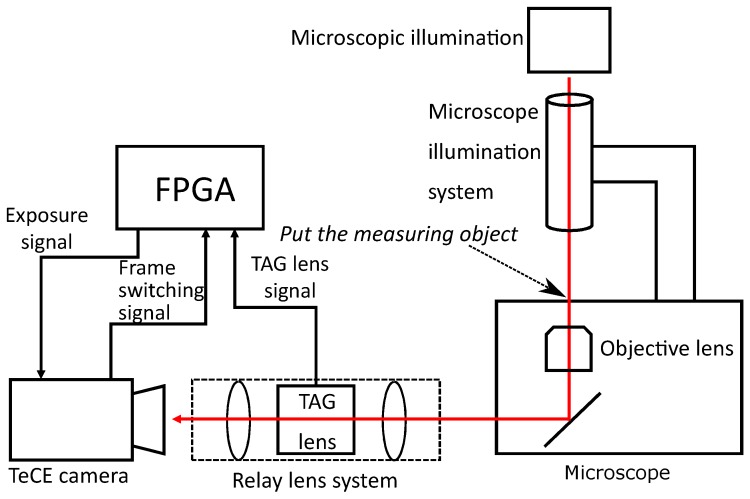
Configuration of experimental system. The field programmable gate array (FPGA) outputs an exposure signal that controls the exposure timing of the TeCE camera. The exposure signal is generated from the TAG lens signal and a frame switching signal. TAG lens signal indicates the trigger signal of the TAG lens.

**Figure 3 sensors-19-01606-f003:**
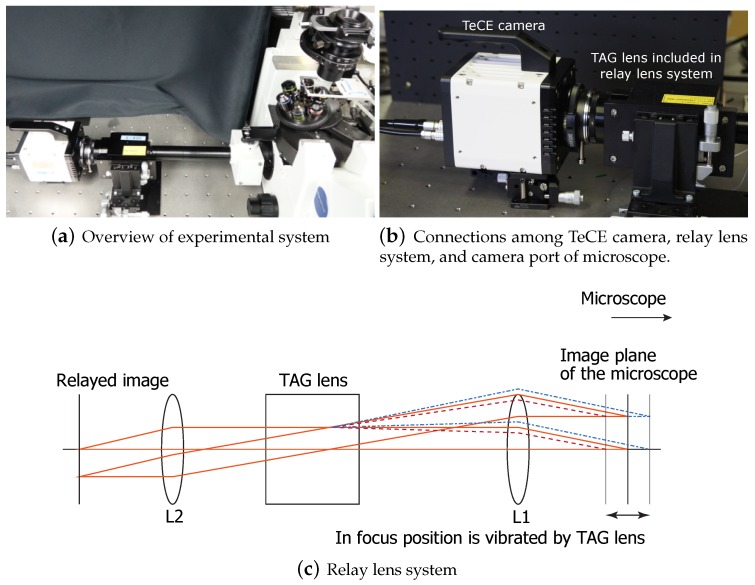
Components of experimental system.

**Figure 4 sensors-19-01606-f004:**

Measurement results of change in brightness depending on exposure times.

**Figure 5 sensors-19-01606-f005:**
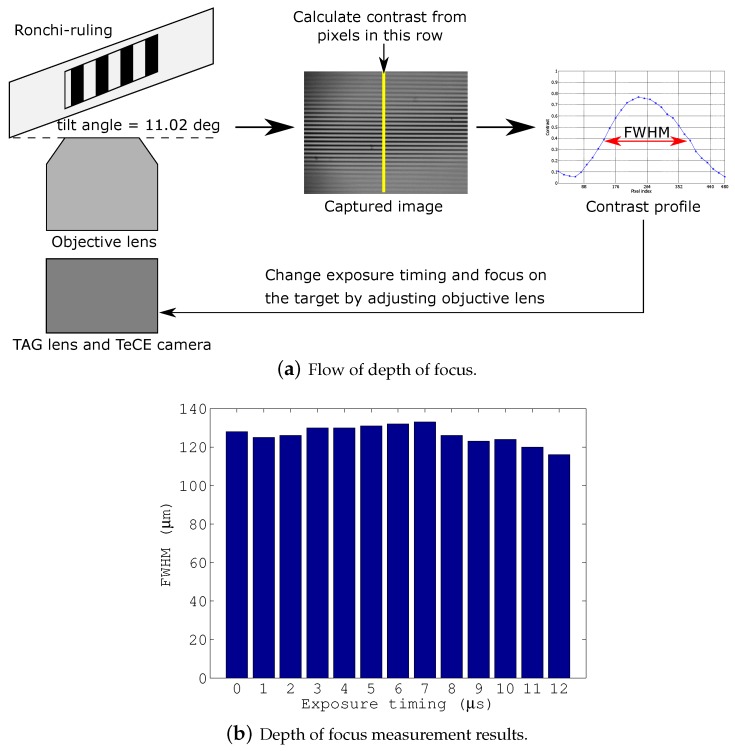
Measurement procedure and results for depth of focus.

**Figure 6 sensors-19-01606-f006:**
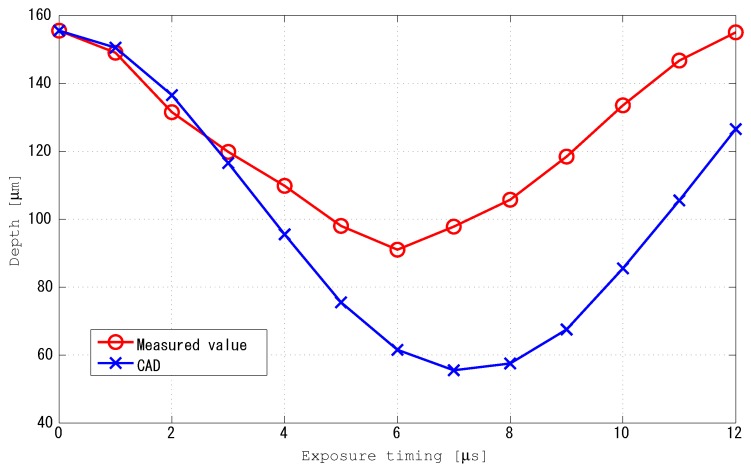
Measurement results for change in focal position.

**Figure 7 sensors-19-01606-f007:**
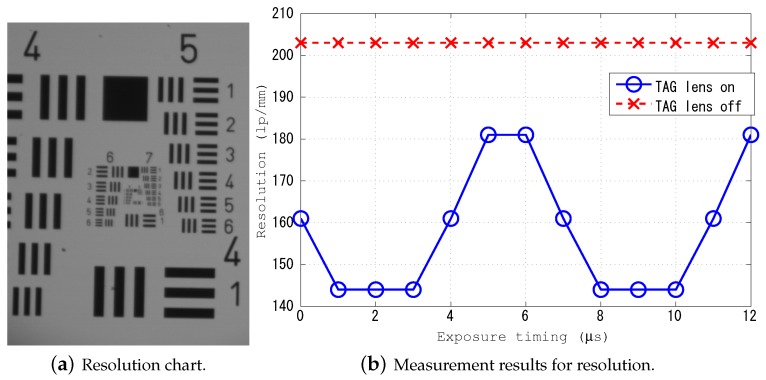
Resolution chart and measurements results.

**Figure 8 sensors-19-01606-f008:**
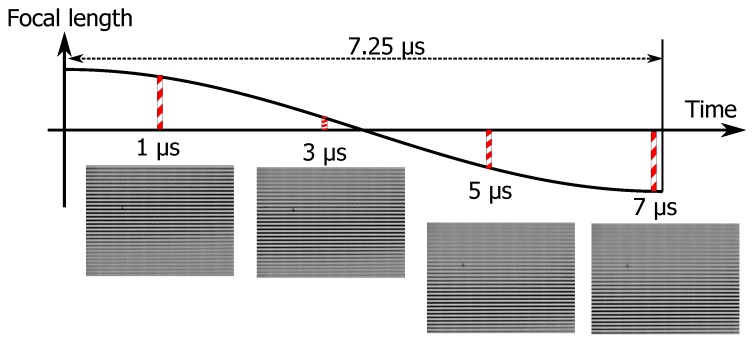
Relation between focal length of TAG lens and captured images.

**Figure 9 sensors-19-01606-f009:**
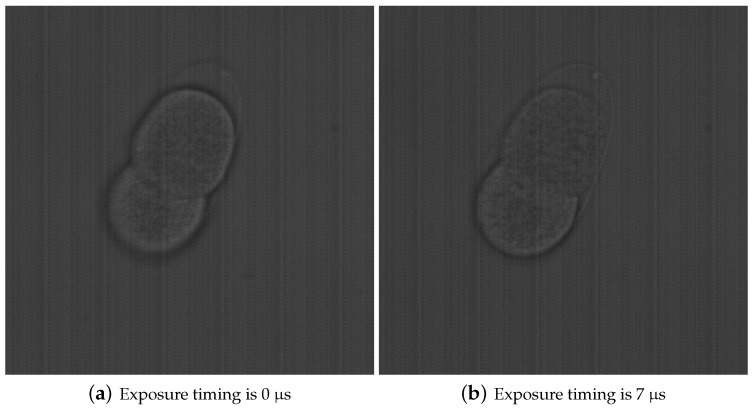
Measured images of two-celled embryo of *C. elegans* by experimental system. The microscope used by this measurement was IX71 (Olympus). The objective lens was UPLANSAPO 60XS2 NA1.3 (Olympus). These images were acquired by bright field observation.

**Figure 10 sensors-19-01606-f010:**
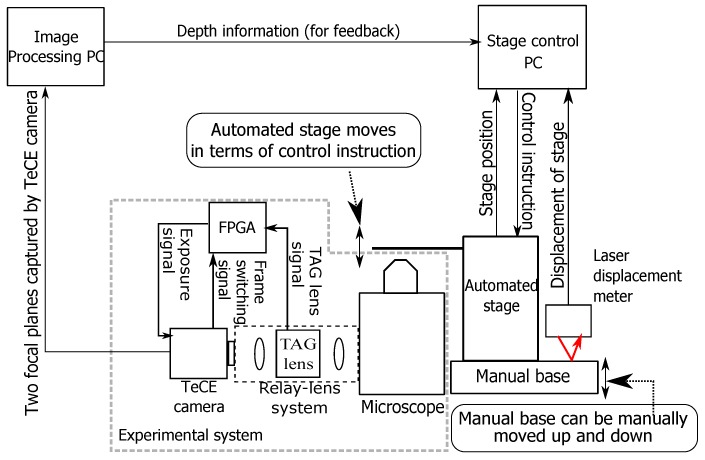
Arrangement of experimental system for focus tracking.

**Figure 11 sensors-19-01606-f011:**
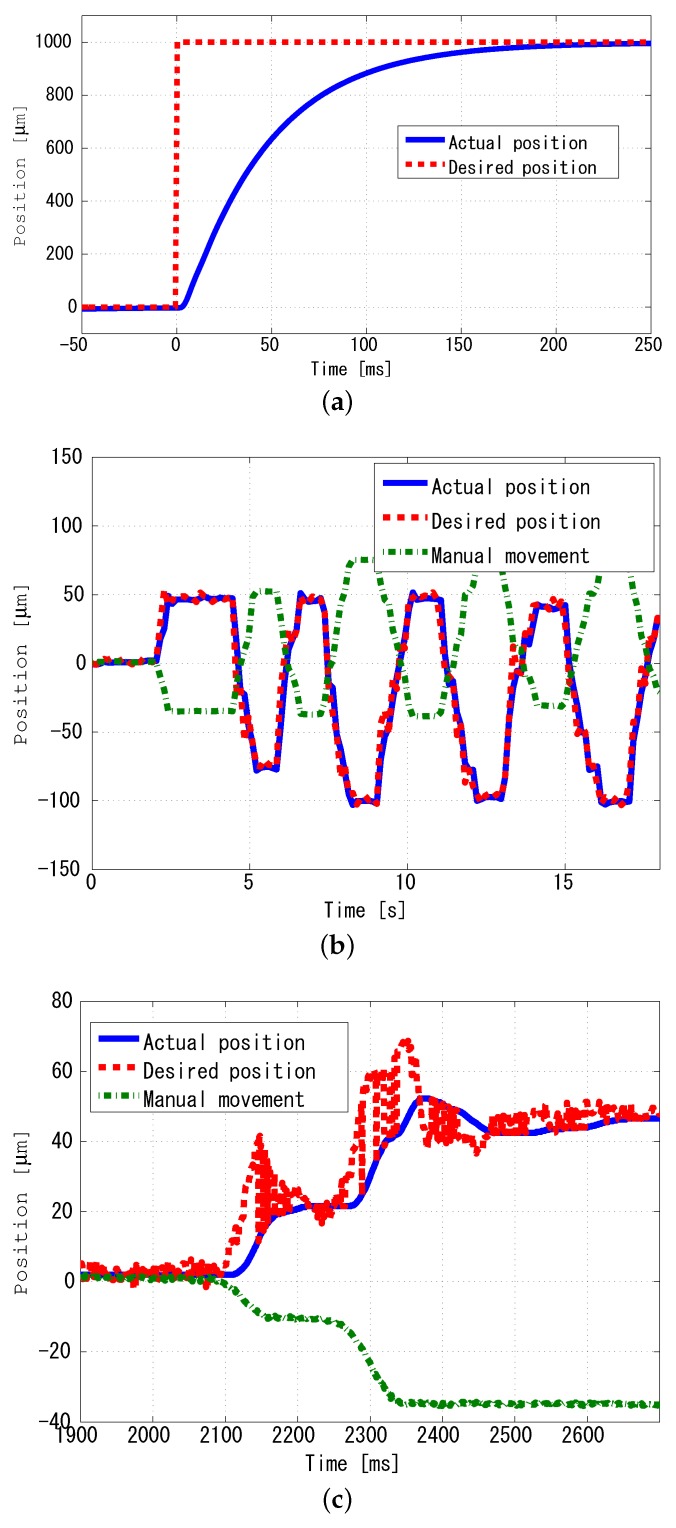
Focus tracking results. (**a**) Step response of the automated stage. In this figure, the actual position shown in the blue solid line shows the current stage position. The desired position shown in the red dashed line is the command value for moving the stage position to the target position. In this figure, the target position is assumed to be 1000 μm; (**b**) Tracking results. Actual position indicates the current position of automated stage. Desired position indicates the command value for automated stage. Manual movement is the displacement of the manual base under the automated stage; (**c**) Enlarged view of (**b**) at 1700 μs to 3000 μs.

**Table 1 sensors-19-01606-t001:** Resolution measurements. This table is used in correspondence with the captured image of the chart shown in Figure 7a.

Element	Group Number									
	−2	−1	0	1	2	3	4	5	6	7
1	0.250	0.500	1.00	2.00	4.00	8.00	16.00	32.0	64.0	128.0
2	0.280	0.561	1.12	2.24	4.49	8.98	17.95	36.0	71.8	144.0
3	0.315	0.630	1.26	2.52	5.04	10.10	20.16	40.3	80.6	161.0
4	0.353	0.707	1.41	2.83	5.66	11.30	22.62	45.3	90.5	181.0
5	0.397	0.793	1.59	3.17	6.35	12.70	25.39	50.8	102.0	203.0
6	0.445	0.891	1.78	3.56	7.13	14.30	28.50	57.0	114.0	228.0

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
