# Peer review of "Fast Volumetric Feedback under Microscope by Temporally Coded Exposure Camera"

_sensors, 2019, doi:10.3390/s19071606_

Round 1

Reviewer 1 Report

The manuscript developed a TeCE camera that can measure  different focus  of a TAG lens at high speed and track the variations of the focus. Focus tracking experiments are excuted to demonstrate the capabilities of the prototype.

Besides the experiments with standard quantities provided by a Ronchi ruling glass slide and a resolution chart, I expect more experiments about measuring real object which is moved at high speed.

Author Response

Thank you for your comment.

I added the measurement result of the real object in our manuscript.

Fig.9 is the added result.

I used the two-celled embryo of C. elegans as a real object.

From this result, it was confirmed that our system could acquire the different focal planes with respect to biological samples.

Could you confirm our manuscript?

Sincerely,

Kazuki Yamato.

Reviewer 2 Report

The manuscript “Fast volumetric feedback under microscope by temporally coded exposure camera” describes an interesting application of TAG lenses for high-speed volumetric measurements under microscopic conditions. The combination of the lenses high-speed focus variation with a temporally coded exposure acquisition presents a novel approach to microscopic high-speed imaging which is shown to be suitable for feedback control in a focus tracking experiment. However, the quality of the manuscript could be improved on several counts. Please see the list below for comments and several issues that were noticed in the manuscript. If the authors revise the submitted manuscript with regard to the points listed here, I can recommend its publication in Sensors.

Page 1, Line 25-26:

§  Unclear formulation: It would improve reader understanding if these systems would be presented as example applications of microscopic 3D-measurements, not that they require them

Page 1, Line 28:

§  Phrasing recommendation: measuring an image acquiring an image

Page 2, Line 63:

§  Which specific optical characteristics were evaluated?

Page 4, Figure 2:

§  It is unclear in the figure where the measurement object is located. Perhaps it could be added to the figure.

§  Since the arrow goes from the FPGA to the camera, the text in the figure description should be “The FPGA outputs an exposure signal […]”

Page 4, Figure 3(c):

§  The text in Fig. 3(c) is too small

§  Switching the light direction so that the microscope is on the right (as in Fig. 2) would improve the readers understanding of the figure

Page 4, Line 115:

§  Correction: “The FPGA outputted an exposure signal […]” (same as the description of Fig. 2)

Page 5, Line 138:

§  There should be some separation between the paragraph title (“Change in brightness depending on exposure time”) and the rest of the text. Without separation it is quite jarring to read because it is read as one sentence.

Page 5, Line 140:

§  Missing space after comma

Page 5, Line 146:

§  Should it not be 14.5 ms? Surely not seconds?

Page 5, Line 149f.:

§  Same issue as in line 138. Missing separation between paragraph title and remaining text.

Page 6, Line 157:

§  Same issue as in line 138.

§  The beginning of the sentence with “In the measurement method […]” should be revised since the grammar seems to be wrong

Page 6, Line 169:

§  Concerning the “slight deviation”: Could you give a concrete value and its significance? What was the uncertainty of the measurement?

Page 7, Line 187:

§  Spelling error: “This reason for this […] The reason for this […]”

Page 7, Line 192:

§  Unclear: Why only two images? Perhaps the explanation given in line 210 could be moved here.

Page 8, Figure 10:

§  The text in the image description is completely messed up and should be corrected.

Page 8, second line after figure:

§  Unclear: What is Reference A? The references at the end are numbered.

Page 9, Line 239 ff.:

§  The conclusion appears too generalized; could you argue your conclusions with some concrete results (such as quantitative results)?

Author Response

Thank you for reviewing our manuscript.

I described the answers to reviewer's comments in a Word file

and uploaded this file.

Could you read this file and confirm our answers?
